# Ciliary and rhabdomeric photoreceptor-cell circuits form a spectral depth gauge in marine zooplankton

Csaba Verasztó[1,2†], Martin Gühmann[1†], Huiyong Jia[3], Vinoth Babu Veedin Rajan[4], Luis A Bezares-Calderón[1,2], Cristina Piñeiro-Lopez[1‡], Nadine Randel[1§], Réza Shahidi[1,2], Nico K Michiels[5], Shozo Yokoyama[3], Kristin Tessmar-Raible[4], Gáspár Jékely[1,2]*

[1]Max Planck Institute for Developmental Biology, Tübingen, Germany; [2]Living Systems Institute, University of Exeter, Exeter, United Kingdom; [3]Department of Biology, Emory University, Atlanta, United States; [4]Max F. Perutz Laboratories, University of Vienna, Vienna, Austria; [5]Department of Biology, University of Tübingen, Tübingen, Germany

**Abstract** Ciliary and rhabdomeric photoreceptor cells represent two main lines of photoreceptor-cell evolution in animals. The two cell types coexist in some animals, however how these cells functionally integrate is unknown. We used connectomics to map synaptic paths between ciliary and rhabdomeric photoreceptors in the planktonic larva of the annelid *Platynereis* and found that ciliary photoreceptors are presynaptic to the rhabdomeric circuit. The behaviors mediated by the ciliary and rhabdomeric cells also interact hierarchically. The ciliary photoreceptors are UV-sensitive and mediate downward swimming in non-directional UV light, a behavior absent in ciliary-opsin knockout larvae. UV avoidance overrides positive phototaxis mediated by the rhabdomeric eyes such that vertical swimming direction is determined by the ratio of blue/UV light. Since this ratio increases with depth, *Platynereis* larvae may use it as a depth gauge during vertical migration. Our results revealed a functional integration of ciliary and rhabdomeric photoreceptor cells in a zooplankton larva.
DOI: https://doi.org/10.7554/eLife.36440.001

*For correspondence:
g.jekely@exeter.ac.uk

[†]These authors contributed equally to this work

Present address: [‡]EMBL Heidelberg, Heidelberg, Germany; [§]Janelia Research Campus, Ashburn, United States

Competing interests: The authors declare that no competing interests exist.

## Introduction

Bilaterian animals have two major photoreceptor cell-types, the rhabdomeric- and the ciliary-type photoreceptor cells (rPRC and cPRC, respectively) (*Arendt, 2003*; *Arendt et al., 2004*; *Eakin, 1979*; *Erclik et al., 2009*). These cells have distinct morphologies and express different classes of opsins (light-sensitive proteins). Rhabdomeric PRCs have apical microvillar specializations (rhabdom) that store the opsin photopigments. The visual photoreceptor cells in most protostome eyes, including the compound eyes of arthropods, the pigment-cup eyes of annelids and the camera or stalk eyes of mollusks, are rhabdomeric and express rhadomeric (r-) opsins (*Arendt et al., 2002*; *Cowman et al., 1986*; *Katagiri et al., 2001*; *Katagiri et al., 1995*; *Ovchinnikov et al., 1988*; *Pollock and Benzer, 1988*; *Randel et al., 2013*). Rhabdomeric PRCs also exist in the pigmented eyespots of both proto-stomes (e.g. annelids, flatworms) and some non-vertebrate deuterostomes (hemichordates, cephalochordates) (*Arendt and Wittbrodt, 2001*; *Braun et al., 2015*; *Nakao, 1964*). In contrast, the visual eyes of vertebrates have cPRCs (rods and cones) where the ciliary (c-) opsin photopigment is stored in specialized ciliary membrane compartments (*Jan and Revel, 1974*; *Nir et al., 1984*). Ciliary PRCs also occur in some invertebrates, where they can be part of pigmented eyespots (e.g. some mollusks and flatworms), in simple eyes (e.g. some nemerteans), or as brain photoreceptors not associated

**eLife digest** The animal kingdom contains many different types of eyes, but all share certain features in common. All detect light using specialized cells called photoreceptors, of which there are two main kinds: ciliary and rhabdomeric. Crustaceans and their relatives, including insects, have rhabdomeric photoreceptors; while animals with backbones, including humans, have ciliary photoreceptors. There are also several groups of animals, mostly sea-dwellers, that inherited both types of photoreceptors from their ancestors that lived millions of years ago. These include the marine ragworm, *Platynereis dumerilii*.

The larvae of *Platynereis* are free-swimming plankton. Each has a transparent brain and six small, pigmented eyes. The eyes contain rhabdomeric photoreceptors. These enable the larvae to detect and swim towards light sources. Yet the larval brain also contains ciliary photoreceptors, the role of which was unknown.

Verasztó, Gühmann et al. now show that ultraviolet light activates ciliary photoreceptors, whereas cyan, or blue-green, light inhibits them. Shining ultraviolet light onto *Platynereis* larvae makes the larvae swim downwards. By contrast, cyan light makes the larvae swim upwards. In the ocean, ultraviolet light is most intense near the surface, while cyan light reaches greater depths. Ciliary photoreceptors thus help *Platynereis* to avoid harmful ultraviolet radiation near the surface. Though if the larvae swim too deep, cyan light inhibits the ciliary photoreceptors and activates the rhabdomeric pigmented eyes. This makes the larvae swim upwards again.

Using high-powered microscopy, Verasztó, Gühmann et al. confirm that neural circuits containing ciliary photoreceptors exchange messages with circuits containing rhabdomeric photoreceptors. This suggests that the two work together to form a depth gauge. By enabling the larvae to swim at a preferred depth, the depth gauge influences where the worms end up as adults. Its discovery should also stimulate new ideas about the evolution of eyes and photoreceptors.
DOI: https://doi.org/10.7554/eLife.36440.002

with pigment cells (e.g. some annelids) (*Arendt et al., 2004*; *Barber et al., 1967*; *Döhren and Bartolomaeus, 2018*; *Randel and Jékely, 2016*).

The class of opsin expressed in a PRC generally correlates with the cell's morphological type, with cPRCs usually expressing c-opsins and rPRCs expressing r-opsins (*Arendt, 2003*; *Arendt et al., 2004*; *Randel et al., 2013*; *Vopalensky et al., 2012*). However some exceptions to this rule are known, such as the occasional coexpression of melanopsin (an r-opsin) with c-opsins or r-opsins with Go- or xenopsins (*Davies et al., 2011*; *Gühmann et al., 2015*; *Vöcking et al., 2017*).

Given their broad phylogenetic distribution and shared opsin expression, both photoreceptor cell types likely coexisted in the last common ancestor of bilaterians. The two cell types still coexist in several marine animals, including cephalochordates, some mollusks, flatworms, and annelids (*Arendt et al., 2004*; *McReynolds and Gorman, 1970*; *Randel and Jékely, 2016*; *Vopalensky et al., 2012*) and form parts of pigmented or non-pigmented light-sensitive structures. Understanding how the two photoreceptor cell types integrate at the functional and circuit levels in these animals will help to clarify the history of eyes and photoreceptor cells.

Here we study the planktonic larva of *Platynereis dumerilii*, a marine annelid that has both photoreceptor cell types. In *Platynereis*, non-pigmented brain cPRCs with ramified cilia express a ciliary type opsin (c-opsin1) (*Arendt et al., 2004*) and coexist with r-opsin-expressing rPRCs that are part of the pigmented visual eyes (adult eyes) and eyespots (*Arendt et al., 2002*; *Jékely et al., 2008*; *Randel et al., 2014*, *2013*). The pigmented larval eyespots and the adult eyes mediate early- and late-stage larval phototaxis, respectively (*Gühmann et al., 2015*; *Jékely et al., 2008*; *Randel et al., 2014*). The mechanism and neuronal circuitry of both early- and late-stage larval phototaxis is well understood. Trochophore larvae (approximately 1–2.5 days post fertilization) have a pair of pigmented eyespots with a rPRC that directly innervates the adjacent ciliary band. This rPRC is cholinergic and expresses r-opsin3 (*Jékely et al., 2008*; *Randel et al., 2013*). When the rPRC is activated during helical swimming, the ciliary beating changes on the illuminated side, so that the larva reorients its trajectory towards the light source. Nectochaete larvae (approximately 3–5 days post fertilization) develop two pairs of adult eyes with several rPRCs coexpressing r-opsin1, r-opsin3 and Go-

opsin (*Gühmann et al., 2015*; *Randel et al., 2013*). These eyes mediate visual phototaxis by comparing the intensity of light on the two sides of the body. The adult eyes and their downstream neuronal circuitry regulate the contraction of the longitudinal muscles of the trunk to steer the larva towards or away from a light source (*Randel et al., 2014*).

The function of cPRCs in *Platynereis* is much less clear. These and the surrounding cells (the 'cPRC region') have been proposed to produce melatonin and to entrain the circadian clock (*Arendt et al., 2004*; *Tosches et al., 2014*). C-opsin1 was recently shown to absorb UV light. This suggests that the c-opsin1-expressing cPRCs mediate circadian entrainment by ambient UV light (*Tsukamoto et al., 2017*). However, the precise function of the cPRCs in *Platynereis* larvae and how they interact with rPRCs is still unknown.

## Results

### Synaptic pathways between rhabdomeric and ciliary photoreceptors

To identify synaptic connections between the cPRC and rPRC circuits, we used a serial-section transmission electron microscopy (ssTEM) dataset spanning the entire body of a 72 hr post fertilization (hpf) *Platynereis* larva (*Randel et al., 2015*). Previously, we reported the synaptic connectome of the rPRCs from the visual eyes and eyespots, (*Randel et al., 2015*, *2014*) and the direct postsynaptic circuit of the four cPRCs with ramified cilia (*Williams et al., 2017a*). Briefly, the glutamatergic rPRCs of the adult eyes connect through three layers of interneurons to cholinergic motoneurons in the ventral head (vMN) that innervate longitudinal trunk muscles. The main premotor interneurons in this circuit are the Schnörkel interneurons (IN$^{sn}$) (*Figure 1C,G*) (*Randel et al., 2015*; *Randel et al., 2014*). The rPRCs of the eyespot synapse directly on the ciliary band and the vMNs (*Figure 1G*) (*Jékely et al., 2008*; *Randel et al., 2015*, *2013*). The four cholinergic cPRCs send axons to the neurosecretory plexus in the anterior nervous system and synapse on four peptidergic/cholinergic RGW interneurons (labelled IN$^{RGW}$, and named after the expression of the RGWamide neuropeptide [*Shahidi et al., 2015*; *Williams et al., 2017*]) and four NOS interneurons (labelled IN$^{NOS}$, and named after the expression of the nitric oxide synthase (NOS) gene; unpublished). The cPRCs also synapse on a group of flask-shaped sensory-neurosecretory neurons that are part of the neurosecretory anterior nervous system. These cells express diverse neuropeptides but have no postsynaptic partners (*Williams et al., 2017*). The RGW cells synapse on two serotonergic cells (Ser-h1) that, together with their postsynaptic partner, the cholinergic MC neuron, are part of the ciliomotor circuitry of the larva (*Verasztó et al., 2017*).

To analyze the possible synaptic integration of the rPRC and cPRC circuits, we searched for all synaptic connections between neurons of these circuits (*Figure 1*) in a synapse-level skeleton reconstruction of all cells in the larval head (2359 cells of which approximately 1230 are neurons; unpublished data). We did a systematic network search in Catmaid (*Schneider-Mizell et al., 2016*) for all possible synaptic paths (two hops) between the rPRC or the cPRC circuit. We identified three sites of contact. First, the RGW interneurons synapse on the Schnörkel premotor interneurons (IN$^{sn}$) (*Figure 1E–G* and *Video 1*). Second, we identified six interneurons (IN$^{preMN}$) that are postsynaptic to the RGW interneurons and presynaptic to the ventral motoneurons (vMNs) of the visual circuit (*Figure 1E–G*). Third, two putative mechanosensory neurons bearing a sensory cilium and a collar of microvilli and located in the median head (MS1 and MS2)(*Bezares-Calderon et al., 2018*) are postsynaptic to the RGW cells and presynaptic to the vMNs and IN$^{pro}$ interneurons of the visual circuit (*Figure 1E–G* and *Video 1*).

Our graph search did not reveal any neurons that were directly postsynaptic to the rPRC circuit (from rPRC to vMN) and presynaptic to any neuron of the cPRC circuit. Thus, the cPRC circuit feeds hierarchically into the visual rPRC circuit (*Figure 1G*). This suggests that the cPRCs could influence phototaxis, a behavior mediated by the rhabdomeric eyes and eyespots (*Gühmann et al., 2015*; *Jékely et al., 2008*; *Randel et al., 2014*).

### Acute UV-violet sensitivity of the ciliary photoreceptors in *Platynereis*

What is the role of cPRCs in larval behavior and how do they influence phototaxis? First, to estimate the light sensitivity of the cPRCs, we reconstructed the morphology with ssTEM and measured the total sensory membrane surface-area of a cPRC (*Figure 2A–D*). Each cPRC has 12–15 basal bodies

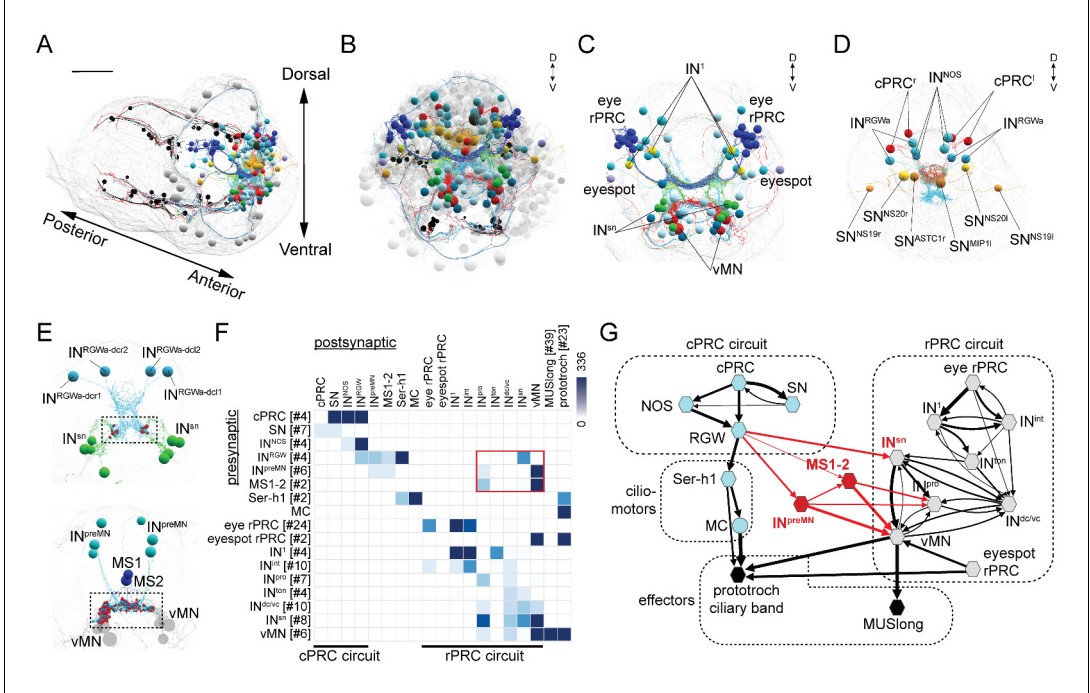

**Figure 1.** Wiring diagram of cPRC and rPRC circuits in the *Platynereis* larval head. (**A**) All cells of the cPRC and rPRC circuits in the larva (in color). The body outline is shown in grey. (**B**) All cells of the cPRC and rPRC circuits (in color), anterior view. All other neurons are shown in grey. Circuits were reconstructed from a whole-body ssTEM volume of a 72 hpf larva. (**C**) All neurons of the rPRC circuit. (**D**) The four cPRCs and all neurons directly postsynaptic to them. (**E**) Connections between the cPRC and rPRC circuits. Top panel: the four RGW interneurons (IN$^{RGW}$) are presynaptic to the IN$^{sn}$ cells. Bottom panel: the six IN$^{preMN}$ cells and the two MS cells are presynaptic to the ventral motoneurons. (**F**) Grouped connectivity matrix of the cPRC and rPRC circuits. The connections from the cPRC to the rPRC circuit are outlined in red. (**G**) Wiring diagram of the cPRC and rPRC circuits. Nodes represent groups of neurons (number indicated in square brackets), arrows represent synaptic connections. Synaptic connections from the cPRC to the rPRC circuit are in red. Edge width is a function of log synaptic count.

DOI: https://doi.org/10.7554/eLife.36440.003

The following source data and figure supplements are available for figure 1:

**Source data 1.** Grouped connectivity matrix of the cPRC and rPRC circuits.
DOI: https://doi.org/10.7554/eLife.36440.006

**Source data 2.** Full ungrouped connectivity matrix of the cPRC and rPRC circuits.
DOI: https://doi.org/10.7554/eLife.36440.007

**Figure supplement 1.** Detailed wiring diagram of the cPRC and rPRC circuits.
DOI: https://doi.org/10.7554/eLife.36440.004

**Figure supplement 2.** Percent of inputs (number of synapses) from the presynaptic cell to the postsynaptic cell, relative to the total number of inputs.
DOI: https://doi.org/10.7554/eLife.36440.005

each with a root and an extensively branched sensory cilium. The branches are supported by single microtubule doublets (*Figure 2—figure supplement 3*). Based on the average diameter and total length of all branches in one cPRC we estimated a membrane area of 276 µm², which is approximately 10 times smaller than the total disk membrane surface area of rat rods (*Mayhew and Astle, 1997*). This suggests that *Platynereis* cPRCs are sensitive enough to mediate acute light sensation.

Next, we expressed *Platynereis* c-opsin1 in COS1 cells, reconstituted it with 11-cis-retinal and purified it. The reconstituted pigment absorbed in the UV range with a λ-max of 384 nm in the dark spectrum and a λ-max of 370 nm in the dark-light difference spectrum (*Figure 2E*), in agreement with a recent report (*Tsukamoto et al., 2017*).

To investigate how the cPRCs respond to light, we did calcium imaging with larvae ubiquitously expressing the calcium sensor GCaMP6s (*Chen et al., 2013*; *Verasztó et al., 2017*). When imaged with a low-intensity 488 nm laser, the cPRCs had a high resting calcium level and the GCaMP6s signal highlighted their sensory cilia. We could thus identify the four cPRCs without stimulation (*Figure 2F*). When the cPRC cilia were locally stimulated (*Figure 2—figure supplement 1*) for 5 min

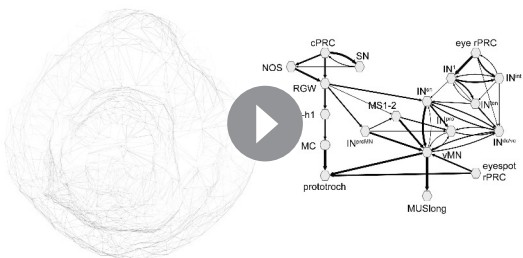

**Video 1.** Wiring diagram of cPRC and rPRC circuits in the *Platynereis* larval head. The anatomy of the reconstructed neurons is shown together with the position of the same cells in the wiring diagram.
DOI: https://doi.org/10.7554/eLife.36440.008

with 405 nm light (140–250 times more photons on the sensory cilia than by the imaging laser), the calcium level dropped transiently at the cPRC somata and then increased strongly. This indicates transient cPRC hyperpolarization followed by depolarization. However, when the cPRCs were stimulated with a 488 nm laser of equal photon flux, they showed prolonged hyperpolarization only, without depolarization (*Figure 2G*). When the 405 nm stimulation was switched off during the hyperpolarization phase (after 20 s), the cPRCs still depolarized afterwards (*Figure 2H*).

## UV-violet-specific activation of neurons postsynaptic to the ciliary photoreceptors

405 nm stimulation of the cPRCs also changed the activity of other neurons in the larval brain. Four neurons followed the activity pattern of the cPRCs (*Figure 2J*; *Figure 2—figure supplement 2*). These cells correspond by position to the four RGW interneurons, which together with the four NOS interneurons are direct postsynaptic targets of the cPRCs (*Figure 1G*)(*Williams et al., 2017a*). Additionally, two flask-shaped sensory neurons (SN$^{early}$) in the middle of the anterior nervous system depolarized upon stimulus onset. Two further sensory cells (SN$^{late}$) flanking the SN$^{early}$ cells depolarized later, in parallel with the rising phase of cPRC activity (*Figure 2I*). These four SN cells correspond by position to four sensory-neurosecretory neurons that are postsynaptic to the cPRCs in the anterior nervous system (SN$^{MIP1l}$, SN$^{ASTC1r}$ and two SN$^{NS20}$ cells)(*Figure 1D*)(*Williams et al., 2017a*). We next compared the responses of the RGW interneurons and the four SN neurons to 405 and 488 nm stimulation of the cPRCs. The RGW interneurons and the SN$^{early}$ sensory neurons only responded to 405 nm stimulation (with opposite sign) but not to 488 nm stimulation. The SN$^{late}$ neurons in some larvae also responded to 488 nm stimulation but these responses were weaker (*Figure 2—figure supplement 2*). Thus, the cPRCs and their postsynaptic neurons respond differentially to violet and blue stimulation, with only violet light inducing cPRC depolarization and consistent changes in the activity of postsynaptic neurons.

## UV-violet avoidance behavior in *Platynereis* larvae

To characterize how *Platynereis* larvae react to UV-violet light, we assayed larval swimming behavior in a vertical column setup. Since *Platynereis* larvae show strong directional phototaxis to a broad spectrum of light (between 380–540 nm)(*Gühmann et al., 2015*; *Jékely et al., 2008*), we illuminated the setup equally from two opposite sides with non-directional UV light so that the larvae could not respond with directional phototaxis (*Figure 3A*).

When the larvae were stimulated with non-directional UV light, they started to swim downward. To characterize the wavelength dependence of this behavior, we assayed larvae in a vertical cuvette and stimulated them with monochromatic light of different wavelengths from two sides. The larvae swam down to UV-violet light (between 340–420 nm) but not to longer wavelengths (>420 nm, *Figure 3B*). This downward swimming UV-avoidance behavior to non-directional UV-violet light has not been previously reported in *Platynereis*. The observations that UV-avoidance can be triggered by non-directional light and has an action spectrum that closely matches the absorption spectrum of c-opsin1 (*Figure 2E*) suggest that the response is mediated by the non-pigmented cPRCs.

If the cPRCs indeed mediate UV-avoidance, then the developmental onset of this response should correlate with the morphological differentiation of cPRCs. To test this, we assayed UV-avoidance (395 nm light from the side) as well as phototaxis (480 nm light from the top) at different larval stages (*Figure 3C*) and correlated the behaviors to photoreceptor differentiation. Phototaxis, but not UV-avoidance was already present at 27 hpf, at a stage when the larval eyespots are already functional (*Jékely et al., 2008*). UV avoidance appeared at 36 hpf, approximately coinciding with the morphological differentiation of cPRCs (after 32 hpf, as judged by the appearance of the

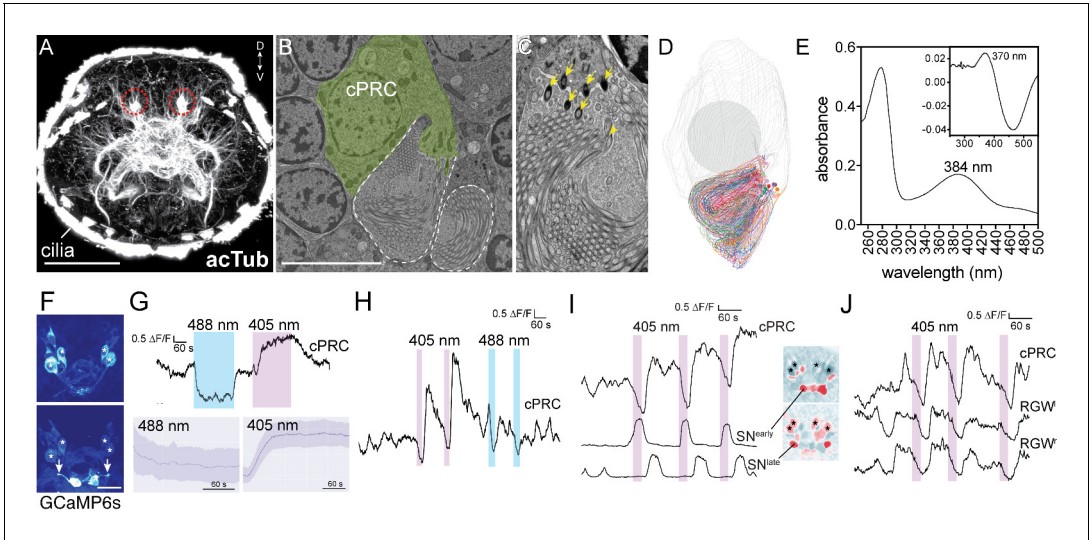

**Figure 2.** Light responses of brain ciliary photoreceptors and their downstream circuitry in *Platynereis* larvae. (**A**) Acetylated tubulin staining of a 72 hpf larva. The ramified sensory cilia of cPRCs are marked with dashed lines. (**B**) TEM image of a section with a cPRC. Cell body in green, sensory cilia outlined in dashed white. (**C**) TEM image of a cPRC with sensory cilia. Yellow arrows mark the basal bodies of a cPRC. (**D**) Serial TEM reconstruction of the sensory cilia of a cPRC. (**E**) Absorption spectrum of purified *Platynereis* c-opsin1. Inset: dark-light difference spectrum. (**F**) Top: high GCaMP6s signal in the cPRCs during imaging conditions. Asterisks mark cPRC nuclei. Bottom: activation of two sensory neurons (SN[early]) upon violet stimulation of cPRCs. (**G**) Top: representative example of cPRC response to prolonged local 488 nm and 405 nm stimulation. The colored boxes show the duration of the stimulation. Bottom: average cPRC response during continuous 488 nm and 405 nm stimulation. Data show mean and s.d. of mean, 488 nm N = 8, 405 nm N> 30. (**H**) Responses of a cPRC to repeated 405 nm and 488 nm (duration: 20 s) stimulation. (**I**) Responses of SN[early] and SN[late] sensory neurons to cPRC 405 nm stimulation. Correlation images are shown for SN[early] and SN[late]. Asterisks mark cPRC nuclei. (**J**) Responses of RGW cells to UV stimulation of a cPRC. Scale bars: (**A**) 50 µm (**B**) 10 µm, (**F**) 20 µm.

DOI: https://doi.org/10.7554/eLife.36440.009

The following source data and figure supplements are available for figure 2:

**Source data 1.** Light and dark-light difference spectrum of *Platynereis* c-opsin1 and calcium imaging traces for panels G-J.
DOI: https://doi.org/10.7554/eLife.36440.014

**Figure supplement 1.** Quantification of stimulus-light intensity during the local stimulation of cPRC cilia.
DOI: https://doi.org/10.7554/eLife.36440.010

**Figure supplement 2.** Calcium imaging in *Platynereis* larvae combined with the stimulation of cPRCs.
DOI: https://doi.org/10.7554/eLife.36440.011

**Figure supplement 2—source data 1.** Source data of *Figure 2—figure supplement 2* panels A-I.
DOI: https://doi.org/10.7554/eLife.36440.012

**Figure supplement 3.** Ultrastructure of cPRCs in a *Platynereis* larva.
DOI: https://doi.org/10.7554/eLife.36440.013

ramified cilia)(*Figure 3C*), but long before the differentiation of the adult eyes (at 72 hpf) (*Fischer et al., 2010*; *Jékely et al., 2008*; *Randel et al., 2014*). Thus, *Platynereis* larvae show UV-violet-light avoidance that is independent of phototaxis, can be induced by non-directional stimulus light, and is likely mediated by the UV-violet-responding non-pigmented cPRCs.

## Antagonistic UV avoidance and phototaxis behaviors form a depth-gauge

To study how UV avoidance interacts with rhabdomeric-eye-mediated phototaxis, we stimulated the larvae in the vertical column with directional monochromatic light from above. When we used 380 nm stimulus light, both early- (41 hpf) and late-stage (3 and 4.5 dpf) larvae first swam upward towards the light (for approximately one minute), and then swam downward (*Figure 3D*, *Video 2*). This upward phase was not observed when the larvae were illuminated uniformly from the side (data not shown). These results indicate that the upward-swimming phase is phototaxis, which is then overwritten by the UV-avoidance response.

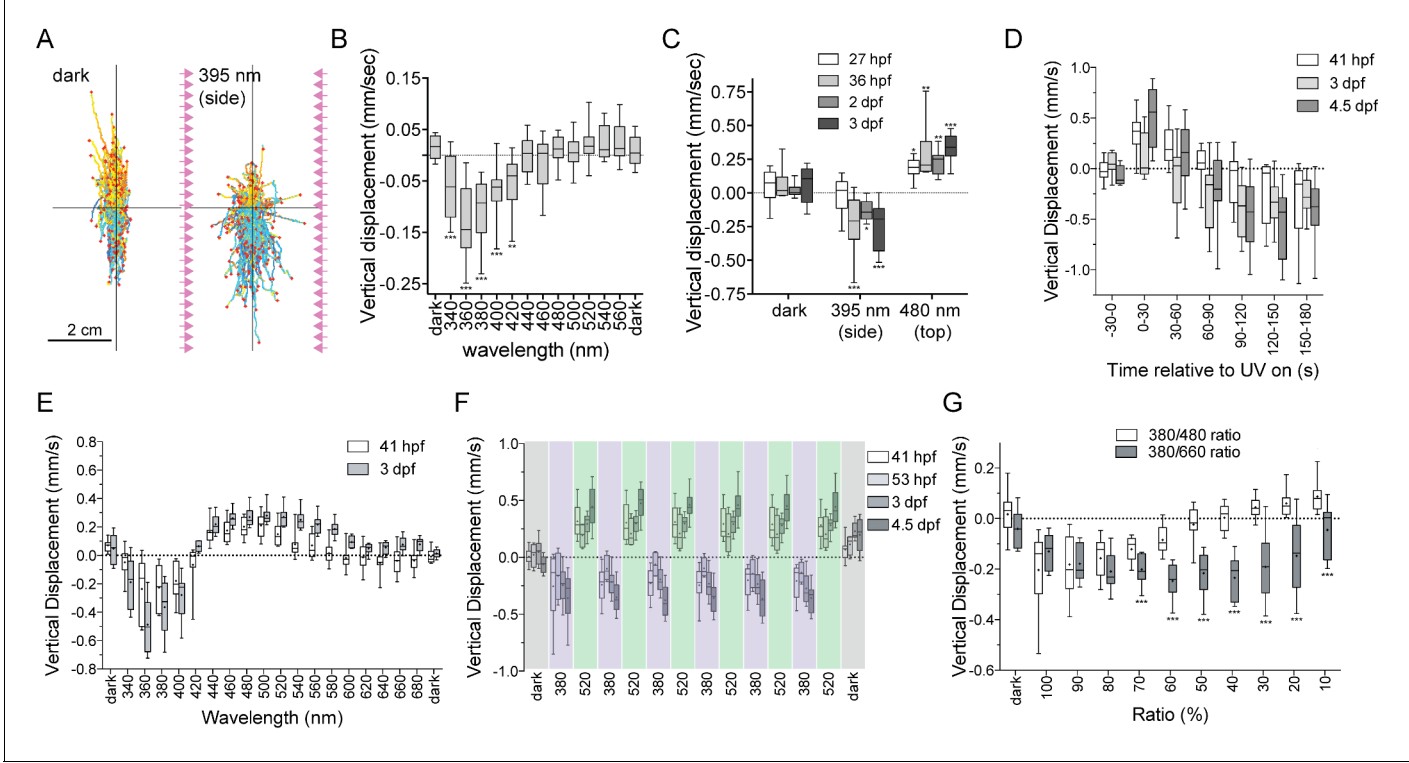

**Figure 3.** UV-violet avoidance and phototaxis form a ratio-chromatic depth gauge in *Platynereis* larvae. (**A**) Larval trajectories recorded in a vertical column in the dark (left) and under illumination with UV light from the side (right). (**B**) Action spectrum of non-directional light avoidance in 48 hpf larvae (n = 20 batches). (**C**) Developmental onset of UV-avoidance behavior and phototaxis in *Platynereis* larvae (n > 7 batches for each stage). (**D**) Time course (30 s bins) of vertical swimming in different larval stages following 380 nm illumination from above (n > 11 batches for each stage). (**E**) Action spectrum of vertical swimming in early- and late-stage larvae, under stimulus light coming from the top of the column. The responses between 1.5–3.5 min after stimulus onset are shown (n > 7 batches for each stage). (**F**) Repeated switching between upward and downward swimming in different larval stages under 380 and 520 nm stimulus light coming from above. The responses between 1.5–4.5 min after stimulus onset are shown (n = 4 batches for 53 hpf, >11 for the other stages). (**G**) Vertical displacement of 3-day-old larvae swimming in a column and stimulated from above with different ratios of 380/480 nm or 380/660 nm monochromatic light (n > 7 batches for each condition). T-tests with Holm-Sidak correction (alpha = 0.05) were used. Significant differences are indicated (*** p-value<0.005).
DOI: https://doi.org/10.7554/eLife.36440.015

The following source data is available for figure 3:

**Source data 1.** Source data of *Figure 3B–G*.
DOI: https://doi.org/10.7554/eLife.36440.016

To test the wavelength-dependence of this effect, we measured larval responses to different wavelengths of directional light coming from the top of the vertical column. We plotted responses 1.5–3.5 min after stimulus onset to focus on the phase when UV-avoidance has potentially overwritten phototaxis (*Figure 3D*). In response to illumination with 340–400 nm light, larvae swam downward after prolonged stimulation. In response to illumination with 440–540 nm light (early-stage) or 440–600 nm light (late-stage), larvae swam upward (*Figure 3E*). 420 nm light did not trigger the vertical displacement of the larvae. The swimming direction could be switched several times by changing the wavelength of the light from 380 nm to 520 nm (*Figure 3F*), demonstrating that this behavioral switch does not habituate even after sustained exposure to light. These results indicate that directional UV-violet light first triggers upward-swimming phototaxis (the pigmented eyes are sensitive in the UV-violet range [*Jékely et al., 2008*; *Gühmann et al., 2015*]), which is then overridden by the downward-swimming UV-avoidance response likely mediated by the cPRCs. Directional blue/cyan light only triggers phototaxis. Importantly, the switching in behavioral response cannot be explained as a wavelength-dependent alternation between positive and negative phototaxis, since

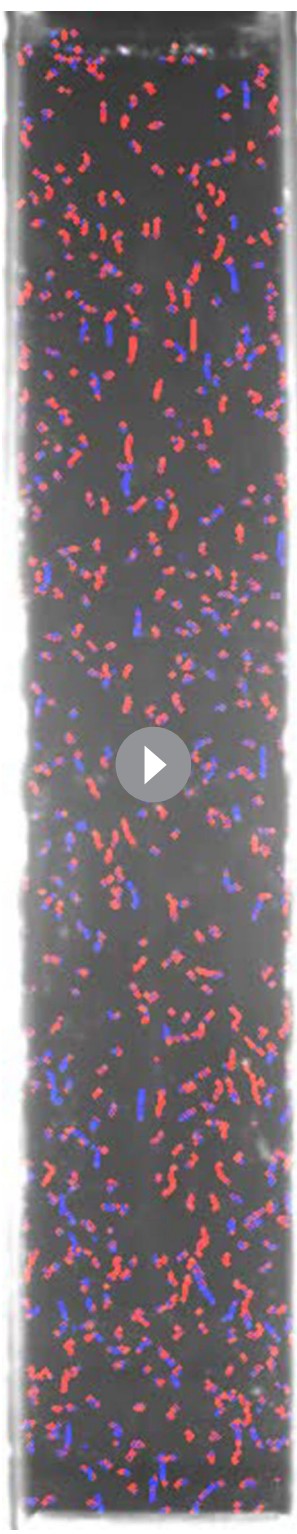

**Video 2.** Behavioral responses of *Platynereis* larvae to 380 nm illumination from above. The magenta square in the top corner marks when the UV stimulus light was switched on. The tracks are color-coded based on heading direction (red, upward; blue, downward). The Video is sped up 2x.

DOI: https://doi.org/10.7554/eLife.36440.017

early-stage larvae are exclusively positively phototactic (*Jékely et al., 2008*)(until 72 hpf [*Randel et al., 2014*]), yet already show the behavioral switch in swimming direction (*Figure 3E,F*).

Next we asked if the antagonistic phototaxis and UV-avoidance behaviors could form a potential mechanism to measure depth by the larvae. Since blue and UV-violet light attenuate differently in seawater, the ratio of blue to UV-violet light increases with depth (field data from the Mediterranean, where our *Platynereis* strain comes from, were shown before [*Gühmann et al., 2015*]). Strong UV-violet light at the ocean's surface is expected to cause larvae to swim downward as an avoidance response, and relatively strong blue light in deeper waters is expected to trigger phototactic upward swimming. Such depth-dependent behavioral switching could form a ratio-chromatic depth-gauge (*Nilsson, 2009*). To test this, we exposed larvae to mixed wavelength light containing different photon ratios of UV (380 nm) and blue (480 nm) light coming from the top of the vertical column. At high 380/480 ratios, larvae swam downward, while at low ratios larvae swam upward. At a 40% 380/480 ratio, larvae remained at a constant average depth (*Figure 3G*), despite being exposed to a directional light stimulus. Mixing the same intensity UV light as before with 660 nm red light (a wavelength to which larvae do not respond phototactically at the intensity used (*Figure 3E*)) did not induce upward swimming at any 380/660 ratio (*Figure 3G*). This experiment shows that a reduction in UV intensity alone does not cause a switch in swimming direction. These results suggest that UV avoidance and phototaxis act antagonistically and cancel each other out under certain wavelength ratios, resulting in no net vertical swimming. This is consistent with the presence of a ratio-chromatic depth gauge in *Platynereis* larvae.

## c-opsin1 knockout larvae have defective UV-violet sensation and avoidance behavior

To test whether c-opsin1 mediates the UV-violet response in *Platynereis* larvae, we used a *c-opsin1 Platynereis* knockout line generated by TALEN-mediated genome editing. The TALENs targeted the third exon of *c-opsin1* and induced an 8 bp deletion (*Figure 4A*). In homozygous *c-opsin1*$^{\Delta 8/\Delta 8}$ larvae, the cPRCs had low resting calcium level and neither hyperpolarized nor depolarized upon 405 nm stimulation (*Figure 4B,C*). Thus, c-opsin1 in the *Platynereis* cPRCs is

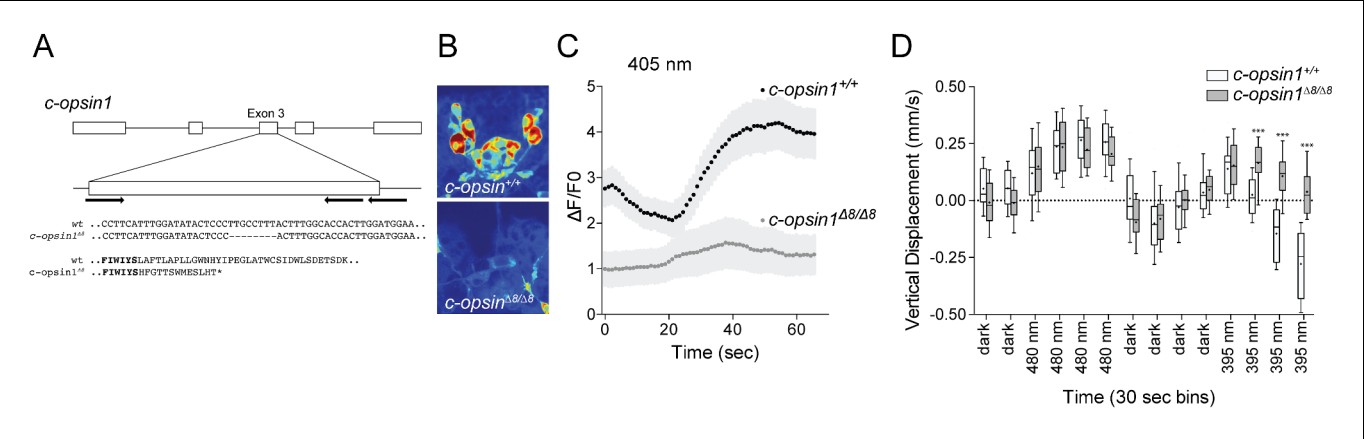

**Figure 4.** *c-opsin1* knockout larvae lack UV-violet responses. (**A**) Schematic of the *Platynereis c-opsin1* gene and the *c-opsin1$^{\Delta8/\Delta8}$* mutation. (**B**) Background GCaMP6s signal during calcium imaging in wild type and *c-opsin1$^{\Delta8/\Delta8}$* mutant larvae. (**C**) Calcium responses to 70 s 405 nm stimulation in wild type and mutant larvae (n > 28 individual larvae for both genotypes). The stimulus light was switched on at 0 s. (**D**) Vertical swimming in wild type and mutant trochophore larvae stimulated with blue (480 nm) and UV (395 nm) light from above (n = 19 for both genotypes). The data are shown in 30 s bins. P-values: ***<0.0001; T-test with Sidak-Bonferroni correction.

DOI: https://doi.org/10.7554/eLife.36440.018

The following source data is available for figure 4:

**Source data 1.** Source data of *Figure 4C*.
DOI: https://doi.org/10.7554/eLife.36440.019
**Source data 2.** Source data of *Figure 4D*.
DOI: https://doi.org/10.7554/eLife.36440.020

required for an elevated resting calcium level in the dark state and for hyperpolarization and subsequent depolarization upon 405 nm exposure. The high resting calcium level in the cPRCs indicates a dark current, a characteristic of vertebrate rods and cones (*Hagins et al., 1970*).

Next, we tested behavioral responses to light in *c-opsin1$^{\Delta8}$* mutant larvae. Similar to wild type larvae, homozygous *c-opsin1$^{\Delta8/\Delta8}$* larvae swim upward (positive phototaxis) in response to 480 nm light. However, *c-opsin1$^{\Delta8/\Delta8}$* larvae have a defective UV-avoidance response. Whereas wild type larvae swim downward after an initial upward-swimming phase, *c-opsin1$^{\Delta8/\Delta8}$* larvae continue to swim upward, showing sustained positive phototaxis in response to UV light (*Figure 4D*). The magnitude of upward swimming diminishes, with close to 0 vertical displacement between 1.5–2 min after UV onset. This indicates that an additional, c-opsin1-independent UV-avoidance mechanism may exist in *Platynereis* larvae (cf. the last bin under 480 and 395 nm stimulation).

These results show that c-opsin1 is a critical mediator of UV-avoidance. The loss of c-opsin1 is expected to disrupt the depth gauge since in *c-opsin1* mutants both cyan and UV light induce sustained positive phototactic behavior which is likely mediated by the pigmented eyes with a broad spectral sensitivity (380–540 nm [*Jékely et al., 2008*]).

## Discussion

Our results are consistent with the presence of a ratio-chromatic depth-gauge in the planktonic larvae of *Platynereis dumerilii*. The depth gauge is formed by the antagonistic interaction of two distinct behaviors, mediated by two distinct types of photoreceptor systems with different spectral sensitivities. The rPRCs mediate phototaxis to a broad range of wavelengths between UV and green light, while the cPRCs mediate UV-violet avoidance. Under UV-violet light, after approximately 30 s the UV-violet avoidance response overrides an initial phototactic response. At a set UV/blue ratio, the two opposing behaviors cancel each other out. Due to the differential attenuation of UV and blue light in seawater, this mechanism could allow the larvae to stay at a certain depth in the ocean.

While the function of the rPRCs in the larval and adult eyes has previously been described (*Jékely et al., 2008*; *Randel et al., 2014*), here we defined, both physiologically and behaviorally, the function of the cPRCs and unraveled how they interact with the rPRC system.

The cPRCs express a UV-absorbing opsin, have a high resting calcium level and react by hyperpolarization and subsequent depolarization to violet light (405 nm). The cPRCs mediate an UV-violet avoidance response characterized by downward swimming. The action spectrum of this behavior matches the absorption spectrum of c-opsin1 and the response is severely impaired in *c-opsin1* knockouts, demonstrating a critical role for c-opsin1 in mediating it. The depolarizing response in the cPRCs shows a similar delayed onset as UV-violet avoidance, suggesting that the behavior is due to cPRC depolarization.

Intriguingly, the depolarization also occurs if the stimulus light is switched off during the hyperpolarizing phase. This implies that cPRC depolarization is not due to direct opsin activation but to a sustained cell-autonomous or non-autonomous signal.

The activation of cPRCs also induces complex differential responses in postsynaptic neurons. The RGW cells follow the activity of cPRCs under violet but not blue illumination, whereas two groups of sensory neurons respond to violet light only (SN$^{early}$) or to both violet and, to a lesser extent, blue light (SN$^{late}$). This implies that different postsynaptic mechanisms operate in the different cPRC targets. The high calcium level in the cPRCs suggests the possibility of the tonic release of a neurotransmitter (probably acetylcholine) in the dark, similar to the tonic release of glutamate by vertebrate photoreceptors (*Heidelberger et al., 2005*). Blue illumination (488 nm) also induces cPRC hyperpolarization, but no subsequent depolarization. This response may be mediated by another opsin expressed in the cPRCs. One candidate is c-opsin2 with a maximal absorption at 490 nm (*Ayers et al., 2018*). One surprising observation is that cPRC hyperpolarization induced by violet or blue light has different downstream consequences. This may be due to the engagement of different signaling cascades by the different opsins.

How cPRC activation and the postsynaptic responses lead to downward swimming during UV-avoidance is unclear. The cPRC circuit connects to the Ser-h1 serotonergic ciliomotor neurons (*Verasztó et al., 2017*), suggesting that the modulation of ciliary activity may contribute to UV avoidance. In addition, we have genetic evidence implicating neuroendocrine volume transmission in UV avoidance. The cPRCs and their direct postsynaptic targets are part of the neurosecretory brain center of the larva (*Williams et al., 2017a*), and knocking out one neuroendocrine signaling component leads to strong defects in UV avoidance (unpublished results).

It also remains to be elucidated how the UV response overrides phototaxis at the neuronal level. We found that the cPRC circuit connects to the circuit of the rhabdomeric eyes via the cholinergic RGW interneurons and their distinct downstream synaptic pathways. The RGW neurons provide inputs to the phototactic circuit at the level of the IN$^{sn}$, IN$^{pro}$ and the vMN cells. The strongest and most direct connection between the cPRC and rPRC circuits is through synapses from the RGW cells to the IN$^{sn}$ neurons. Cholinergic input from the RGW cells to these premotor interneurons may antagonize phototaxis. Other sensory inputs may also tune phototaxis via the RGW-IN$^{sn}$ pathway since neuropeptides derived from sensory-neurosecretory cells can influence RGW neuron activity (*Williams et al., 2017*). The RGW neurons also connect to the putatively mechanosensory MS cells which strongly synapse on the vMNs. UV light may thus tune mechanosensation that may in turn modulate phototaxis.

One outstanding question in eye evolution is why did the ciliary and rhabdomeric PRC types originally evolve (*Nilsson, 2013*; *2009*)? Our findings suggest that the two types may have evolved antagonistic functions early in evolution. Invertebrates then elaborated on the rhabdomeric, and vertebrates on the ciliary type (*Arendt et al., 2004*). According to one hypothesis, as the brain cPRCs were recruited for vision in the vertebrate lineage, the rPRCs evolved into the retinal ganglion cells (*Arendt, 2008*; *Arendt, 2003*). This scenario is supported by the expression of melanopsin, an r-opsin ortholog, in the intrinsically photosensitive retinal ganglion cells (*Hattar et al., 2003*; *2002*; *Koyanagi et al., 2005*; *Lucas et al., 2001*; *Panda et al., 2003*). Although we need more comparative data to test this model, we hypothesize that the cell-type mosaic of the vertebrate retina may have originated from the hierarchical integration of distinct cPRC and rPRC circuits mediating antagonistic behaviors, as observed in *Platynereis* larvae.

# Materials and methods

## Key resources table

| Reagent type (species) or resource | Designation | Source or reference | Identifiers | Additional information |
|---|---|---|---|---|
| Strain (*Platynereis dumerilii*) | *c-opsin1*$^{\Delta8/\Delta8}$ knockout | This paper | | Knockout generated by TALEN-induced gene editing |
| Cell line (*Cercopithecus aethiops*) | COS1 | | RRID:CVCL_0223 | ATCC® CRL1650™ |
| Transfected construct (*Platynereis dumerilii*) | pMT5-c-opsin1 | This paper | | Expression plasmid |
| Biological sample (*Platynereis dumerilii*) | Wild type Tübingen strain | Other | | Jékely lab strain (Tübingen, Exeter) |
| Sequence-based reagent | cops1_F1 | This paper | | GACCTACCTCCCAAATAAGTGATG |
| Sequence-based reagent | cops1_R1 | This paper | | CTGTGGCGGACGAGGCTGGCC |
| Sequence-based reagent | cops1_F2 | This paper | | GACCCGTAGCAGCCACTCCC |
| Sequence-based reagent | cops1_R2 | This paper | | GGTCTGGGAGCCCTGATGACTC |
| Sequence-based reagent | cops1_F3 | This paper | | CGCTGGAACTTACCTTTCTGAC |
| Sequence-based reagent | cops1_R3 | This paper | | GCCTTCATTTGGATATACTCCC |
| Sequence-based reagent | cops1_F4 | This paper | | CACCTGCTTATTCATGAAGACG |
| Sequence-based reagent | cops1_R4 | This paper | | GGTGGCTAAAACTGGTGGAAG |
| Sequence-based reagent | cops1_F5 | This paper | | GCTGGCAACTTATGTAAACAAACAG |
| Sequence-based reagent | cops1_R5 | This paper | | CTTTTTTCATTGCAGTTCCGAAG |
| Sequence-based reagent | TAL_F1 | This paper | | TTGGCGTCGGCAAACAGTGG |
| Sequence-based reagent | TAL_R2 | This paper | | GGCGACGAGGTGGTCGTTGG |
| Sequence-based reagent | cops1_TAL_R1 | This paper | | GCCTTCATTTGGATATACTCCCTTG |
| Sequence-based reagent | cops1_TAL_L2 | This paper | | CGCTGGAACTTACCTTTCTGAC |
| Sequence-based reagent | cops1_del8m_F | This paper | | ATACTCCCTTGCCTTTACCACTT |
| Sequence-based reagent | cops1_del8w_F | This paper | | TATACTCCCTTGCCTTTACTTTGG |
| Sequence-based reagent | cops1_com_R | This paper | | CAAGTTTTGTAAGTGAAATTGCATCC |
| Sequence-based reagent | cops1_ del8_2F | This paper | | AGCCTTCATTTGGATATACTCCC |
| Sequence-based reagent | cops1_del8_2R | This paper | | TTATAAACGATGGAACTTACCTTTCTG |
| Sequence-based reagent | cops1_F1 | This paper | | GACCTACCTCCCAAATAAGTGATG |
| Sequence-based reagent | cops1_R1 | This paper | | CTGTGGCGGACGAGGCTGGCC |
| Sequence-based reagent | cops1_F2 | This paper | | GACCCGTAGCAGCCACTCCC |
| Sequence-based reagent | cops1_R2 | This paper | | GGTCTGGGAGCCCTGATGACTC |
| Sequence-based reagent | cops1_F3 | This paper | | CGCTGGAACTTACCTTTCTGAC |
| Sequence-based reagent | cops1_R3 | This paper | | GCCTTCATTTGGATATACTCCC |
| Sequence-based reagent | cops1_F4 | This paper | | CACCTGCTTATTCATGAAGACG |
| sequence-based reagent | cops1_R4 | This paper | | GGTGGCTAAAACTGGTGGAAG |
| Sequence-based reagent | cops1_F5 | This paper | | GCTGGCAACTTATGTAAACAAACAG |
| Sequence-based reagent | cops1_R5 | This paper | | CTTTTTTCATTGCAGTTCCGAAG |

*Continued on next page*

*Continued*

| Reagent type (species) or resource | Designation | Source or reference | Identifiers | Additional information |
|---|---|---|---|---|
| Sequence-based reagent | TAL_F1 | This paper | | TTGGCGTCGGCAAACAGTGG |
| Sequence-based reagent | TAL_R2 | This paper | | GGCGACGAGGTGGTCGTTGG |
| Sequence-based reagent | cops1_TAL_R1 | This paper | | GCCTTCATTTGGATATACTCCCTTG |
| Sequence-based reagent | cops1_TAL_L2 | This paper | | CGCTGGAACTTACCTTTCTGAC |
| Sequence-based reagent | cops1_del8m_F | This paper | | ATACTCCCTTGCCTTTACCACTT |
| Sequence-based reagent | cops1_del8w_F | This paper | | TATACTCCCTTGCCTTTACTTTGG |
| Sequence-based reagent | cops1_com_R | This paper | | CAAGTTTTGTAAGTGAAATTGCATCC |
| Sequence-based reagent | cops1_ del8_2F | This paper | | AGCCTTCATTTGGATATACTCCC |
| Sequence-based reagent | cops1_del8_2R | This paper | | TTATAAACGATGGAACTTACCTTTCTG |
| Sequence-based reagent | cops1_nest_seq | This paper | | ATGAGACCATACGAAACCAC |
| Commercial assay or kit | Phusion Human Specimen Direct PCR Kit | Thermofisher | | |
| Commercial assay or kit | mMESSAGE mMACHINE Sp6 kit | Thermofisher | | |
| Commercial assay or kit | Golden Gate TAL Effector Kit 2.0, | Addgene 1000000024 | | |
| Software, algorithm | Fiji | PMID: 22743772 | RRID:SCR_002285 | |
| Software, algorithm | perl and Fiji scripts for tracking | https://github.com/JekelyLab/Veraszto_et_al_2018 (copy archived at https://github.com/elifesciences-publications/Veraszto_et_al_2018) | 0000d2a | |

## *Platynereis dumerilii* culture

Larvae of *Platynereis dumerilii* were cultured at 18°C in a 16 hr light 8 hr dark cycle until experiments. Larvae were raised to sexual maturity according to established breeding procedures (*Fischer and Dorresteijn, 2004*; *Hauenschild and Fischer, 1969*).

## Estimation of cPRC sensory membrane surface

We used serial-sectioning transmission electron microscopy to analyze cPRC sensory morphology (*Randel et al., 2015*). Each cPRC has 12–15 basal bodies, each basal body gives rise to one cilium that branches close to its base to 3–9 branches. Each branch is supported by at least one microtubule doublet. The branches have an average diameter of 130 nm (st.dev. 19 nm) and a total length of 677 µm (measured in one cPRC). This represents a total membrane surface area of approximately 276 µm$^2$.

## In vitro absorption spectrum of c-opsin1

Recombinant c-opsin1 was purified and analyzed following (*Yokoyama, 2000*). Full-length *Platynereis c-opsin1* was amplified with primers that introduced EcoRI, Kozak and SalI sequences, and cloned into the expression plasmid pMT5. C-opsin1 was expressed in COS1 cells and incubated with 11-cis-retinal (a gift from Dr. Rosalie K. Crouch at Storm Eye Institute) to regenerate the photopigment.

The pigment was purified with immobilized 1D4 (The Culture Center, Minneapolis, MN) in buffer W1 (50 mM N-(2-hydroxyethyl) piperazine-N'−2-ethanesulfonic acid (HEPES) (pH 6.6), 140 mM NaCl, 3 mM MgCl2, 20% (w/v) glycerol and 0.1% dodecyl maltoside). The UV/VIS spectrum of the pigment was recorded at 20°C with a Hitachi U-3000 dual beam spectrophotometer. The pigment was bleached for 3 min with a 60 W standard light bulb equipped with a Kodak Wratten #3 filter at a distance of 20 cm. COS1 cells (ATCC CRL1650), established from the kidney cells of the African green monkey (*Cercopithecus aethiops*), were authenticated by the American Type Culture Collection (Manassas, VA) with the COI assay. The mycoplasma contamination test was negative.

## Photostimulation

We used a variety of light sources for photostimulation, depending on the experimental setup. In the calcium imaging experiments, we used the laser lines available in our Olympus FV1200 confocal microscope (405 and 488 nm; Showa Optronics, Tokyo). The lasers were operating in continuous mode. The power of the lasers was measured with a microscope slide power sensor (S170C; Thorlabs, Newton, USA). The typical power for stimulation was 5.59 µwatts for the 405 nm laser and 4.62 µwatts for the 488 nm laser. These values correspond to 1.1e + 13 photons/sec. For behavioral assays, we used either UV LEDs (395 nm peak wavelength) or a monochromator (Polychrome II, Till Photonics) for which we quantified photon irradiances across the spectrum ($3-4 \times 10^{18}$ photons/sec/$m^2$) (*Gühmann et al., 2015*). We refer to color according to these wavelength ranges: UV <400 nm, violet 400–450 nm, blue 450–490 nm (450–460 royal blue), cyan 490–520 nm.

## Calcium imaging

For calcium imaging, 36–52 hpf larvae were used. Experiments were conducted at room temperature in filtered natural seawater. Larvae were immobilized between a slide and a coverslip spaced with adhesive tape. GCaMP6s mRNA (1 mg/µl) was injected into zygotes as described previously (*Randel et al., 2014*). Larvae were imaged on an Olympus FV1200 microscope (with a UPLSAPO 60X water-immersion objective, NA 1.2) with a frame rate of 1.25/sec and an image size of 254 × 254 pixels. The larvae were stimulated in a region of interest (a circle with 18–24 pixel diameter) with continuous 405 nm or 488 nm lasers controlled by the SIM scanner of the Olympus FV12000 confocal microscope while scanning. The imaging laser had a similar intensity than the stimulus laser but covered an area that was 140–250 times larger than the stimulus ROI.

## Calcium-imaging data analysis and image registration

The calcium-imaging movies were analyzed with Fiji (*Schindelin et al., 2012*) (RRID:SCR_002285) and a custom Python script, as described previously (*Gühmann et al., 2015*), with the following modifications. The movies were motion-corrected in Fiji with moco (*Dubbs et al., 2016*) and Descriptor-based registration (https://github.com/fiji/Descriptor_based_registration). The data are shown as ΔF/F0. For the calculation of the normalized ΔF/F0 with a time-dependent baseline, F0 was set as the average of an area of the brain with no calcium activity to normalize for the additional lasers (405 nm and 488 nm) and potential artefacts from the microscope's detector. Spatial correlation analyses of neuron activities were done in Fiji and Python as previously described (*Verasztó et al., 2017*). The ROI was manually defined and was correlated with every pixel of the time series. Finally, a single image was created with the Pearson correlation coefficients and a [−1, 1] heatmap plot with two colors. Scripts are available at (*Gühmann and Verasztó, 2018*; copy archived at https://github.com/elifesciences-publications/Veraszto_et_al_2018).

## Vertical column setup for measuring photoresponses

Photoresponses of larvae of different ages were assayed in a vertical Plexiglas column (31 mm x 10 mm x 160 mm water height). The column was illuminated from above with light from a monochromator (Polychrome II, Till Photonics). The monochromator was controlled by AxioVision 4.8.2.0 (Carl Zeiss MicroImaging GmbH) via analog voltage or by a custom written Java program via the serial port. The light passed a collimator lens (LAG-65.0–53.0 C with MgF2 Coating, CVI Melles Griot). The column was illuminated from both sides with light-emitting diodes (LEDs). The LEDs on each side were grouped into two strips. One strip contained UV (395 nm) LEDs (SMB1W-395, Roithner Lasertechnik) and the other infrared (810 nm) LEDs (SMB1W-810NR-I, Roithner Lasertechnik). The UV

LEDs were run at 4 V to stimulate the larvae in the column from the side. The infrared LEDs were run at 8 V (overvoltage) to illuminate the larvae for the camera (DMK 22BUC03, The Imaging Source), which recorded videos at 15 frames per second and was controlled by IC Capture (The Imaging Source).

### Non-directional UV-light assay

27-hour-old, 36-hour-old, 2-day-old, and 3-day-old *Platynereis dumerilii* larvae were stimulated in the column with UV (395 nm) light from the LEDs on the side. Afterwards, the larvae were stimulated with monochromatic blue (480 nm) light coming from above to assay for phototaxis. Each stimulus lasted 4 min. The LEDs were controlled manually and the monochromator (Polychrome II, Till Photonics) was controlled via AxioVision.

### Comparing behavior of wildtype and *c-opsin1*-knockout larvae

To compare the behavior of wildtype and *c-opsin1*-knockout larvae in the vertical column, we tested individual batches of larvae. We distributed the larvae in the vertical column by mixing and dark adapted them for 5 min. The larvae were recorded for 1 min in the dark followed by exposure to collimated blue (480 nm) light from the top of the column for 2 min, then 2 min darkness, and finally collimated UV (395 nm) light from above for 2 min. Stimulus light was provided by a monochromator (Polychrome II, Till Photonics).

### UV-green-light switching assay

Early and late-stage *Platynereis dumerilii* larvae were assayed in the vertical columns. The larvae were stimulated six times alternatively with UV (380 nm) and green (520 nm) light. Each stimulus lasted 4.5 min. The last 3 min within each stimulus were analyzed for vertical displacement of the larvae. The light was provided by a monochromator (Polychrome II, Till Photonics), which was controlled by AxioVision.

### Action spectrum of vertical swimming

2-day-old and 3-day-old *Platynereis dumerilii* larvae were assayed in vertical columns. The larvae were stimulated with monochromatic light from above between 340 nm and 480 nm in 20 nm steps. Between the stimuli, additional 520 nm stimuli were introduced to avoid the accumulation of the larvae at the bottom after UV treatment. The larvae were also stimulated with monochromatic light from above between 400 nm and 680 nm in 20 nm steps. Between these stimuli, additional 400 nm stimuli were introduced to avoid the accumulation of the larvae at the top due to phototaxis. Each stimulus lasted 3.5 min. The last 2 min of each stimulus were analyzed for vertical displacement of the larvae. The light was provided by a monochromator (Polychrome II, Till Photonics), which was controlled by AxioVision.

### Ratio-metric assay

3-day-old *Platynereis dumerilii* larvae were stimulated with UV-blue (380 nm, 480 nm) or UV-red (380 nm, 660 nm) mixed light from above. The larvae were mixed to be uniformly distributed in the column and dark adapted for 5 min. In each step, 10% UV-light was replaced by blue or red light. Each step was followed by a blue (480 nm) light stimulus to avoid the accumulation of the larvae at the bottom after UV treatment. Different ratios were created by rapidly switching between wavelengths within 500 ms periods (e.g., for a 10% UV 90% blue ratio we provided UV-light for 50 ms followed by blue light for 450 ms). Each stimulus condition lasted 4 min. The monochromator was controlled by a custom Java program via the serial port.

### Vertical cuvette photoresponse assay

2-day-old *Platynereis dumerilii* larvae were assayed in a vertical cuvette of 10 mm x 10 mm x 42 mm (L x W x H). The larvae were kept at 18°C and had been exposed to daylight before the experiment. The larvae were illuminated with a monochromator (Polychrome II, Till Photonics) from one side of the cuvette. A mirror (PFSQ20-03-F01, Thorlabs) placed at the opposite side reflected the light to provide near-uniform bilateral illumination. The light passed a diffuser (Ø1' 20° Circle Pattern Diffuser, ED1-C20; Thorlabs) and a collimating lens (LAG-65.0–53.0 C with MgF2 Coating, CVI Melles

Griot) before it hit the cuvette. The cuvette was illuminated with infrared (850 nm) light-emitting diodes (LEDs) (L2 × 2-I3CA, Roithner Lasertechnik) from the side. The LEDs were run at 6.0 V. The larvae were stimulated with light from 340 nm to 560 nm in 20 nm steps. Each step lasted 1 min. The steps were separated by 1 min darkness, so that the larvae could redistribute after each stimulus. The larvae were recorded at 16 frames per second with a DMK 23GP031 camera (The Imaging Source) controlled by IC Capture. The camera was equipped with a macro objective (Macro Zoomatar 1:4/50–125 mm, Zoomar Muenchen). It was mounted with a close-up lens (+2 52 mm, Dörr close-up lens set 368052). The larvae were tracked and their vertical displacement was analyzed during the last 45 s of each stimulus period. Scripts are available at (*Gühmann and Verasztó, 2018*).

## Generation of c-opsin1 knockouts

The genomic region of *c-opsin1* was amplified to screen for putative size-polymorphic alleles or single nucleotide polymorphisms (SNPs) from different *Platynereis* strains (PIN and VIO strains) with the following screening primers: cops1_F1/R1, cops1_F2/R2, cops1_F3/R3, cops1_F4/R4 and cops1_F5/R5 (For a detailed protocol for SNP screening see: (*Bannister et al., 2014*).

*c-opsin1* TALEN pairs were designed in several non-polymorphic exon regions with the TALE-NT prediction tool (TAL Effector Nucleotide Targeter 2.0; https://tale-nt.cac.cornell.edu/) (*Doyle et al., 2012*). The in silico predictions were done with customized design conditions - 15 left/right Repeat Variable Diresidue (RVD) length, 15–25 bp spacer length, G substitute by NN RVD and presence of a restriction enzyme site within the spacer region. The predicted *c-opsin1* TALENs were constructed in vitro with the Golden Gate assembly protocol (Golden Gate TAL Effector Kit 2.0, Addgene #1000000024) (*Cermak et al., 2011*). The final TALEN repeats were cloned to heterodimeric FokI expression plasmids pCS2TAL3-DD for left TALEN array and pCS2TAL3-RR for right TALEN array (*Dahlem et al., 2012*). All cloned TALEN plasmids were sequence-verified with the TAL_F1 and TAL_R2 primers. *c-opsin1* TALEN mRNA for each array was generated by linearizing the corresponding plasmid (NotI) and transcribing it in vitro with the mMESSAGE mMACHINE Sp6 kit (Thermofisher). 200 ng/µl/TALEN mRNA was microinjected into *Platynereis* zygotes (*Backfisch et al., 2013*) and screened for TALEN-induced mutations (*Bannister et al., 2014*) with the PCR primers cops1_TAL_R1/cops1_TAL_L2 followed by restriction digest by the BanI enzyme for the TAL_pair3 and MluCI for the TAL_pair4. The injected embryos were raised to maintain knockout cultures.

## *c-opsin1* sequencing and genotyping

For genotyping of the *c-opsin1* locus, genomic DNA was isolated from single larvae, groups of 6–20 larvae, or from the tails of adult worms. The DNA was amplified by PCR with the cops1_del8m_F/cops1_com_R or the cops1_ del8_2F/cops1_ del8_2R primers with the dilution protocol of the Phusion Human Specimen Direct PCR Kit (Thermo Scientific). The PCR product was sequenced directly with a nested sequencing primer cops1_ nest_seq. A mixture of wild-type and deletion alleles in a sample gave double peaks in the sequencing chromatograms, with the relative height of the double peaks reflecting the relative allele ratio in the sample.

## Acknowledgements

We thank Elizabeth A Williams for comments. The research was supported by a grant from the DFG - Deutsche Forschungsgemeinschaft (Reference no. JE 777/3–1). SY was supported by the National Institutes of Health (R01EY016400) and Emory University. KTR is supported by grants from the University of Vienna (research platform "Rhythms of Life"), the FWF (http://www.fwf.ac.at/en/) research project grant (#P28970), and the European Council under the European Community's Seventh Framework Programme (FP7/2007-2013) ERC Grant Agreement 337011.

## Additional information

### Funding

| Funder | Grant reference number | Author |
| --- | --- | --- |
| National Institutes of Health | | Vinoth Babu Veedin Rajan Shozo Yokoyama |

| | | |
|---|---|---|
| Deutsche Forschungsge-meinschaft | JE 777/3–1 | Luis A Bezares-Calderón |
| University of Vienna Marine Rhythms of Life | | Kristin Tessmar-Raible |
| FP7 Ideas: European Research Council | Grant Agreement 33701 | Kristin Tessmar-Raible |
| Austrian Science Fund | P28970 | Kristin Tessmar-Raible |
| Max-Planck-Gesellschaft | Open-access funding | Gáspár Jékely |

The funders had no role in study design, data collection and interpretation, or the decision to submit the work for publication.

### Author contributions
Csaba Verasztó, Martin Gühmann, Data curation, Software, Formal analysis, Validation, Investigation, Visualization, Methodology, Writing—review and editing; Huiyong Jia, Vinoth Babu Veedin Rajan, Cristina Piñeiro-Lopez, Nico K Michiels, Shozo Yokoyama, Investigation, Methodology; Luis A Bezares-Calderón, Réza Shahidi, Data curation, Investigation; Nadine Randel, Data curation, Formal analysis, Investigation, Visualization; Kristin Tessmar-Raible, Investigation, Methodology, Writing—review and editing; Gáspár Jékely, Conceptualization, Data curation, Formal analysis, Supervision, Funding acquisition, Validation, Visualization, Methodology, Writing—original draft, Project administration, Writing—review and editing

### Author ORCIDs
Csaba Verasztó https://orcid.org/0000-0001-6295-7148
Martin Gühmann http://orcid.org/0000-0002-4330-0754
Vinoth Babu Veedin Rajan http://orcid.org/0000-0002-2430-7395
Nadine Randel https://orcid.org/0000-0002-7817-4137
Gáspár Jékely http://orcid.org/0000-0001-8496-9836

### Decision letter and Author response
Decision letter https://doi.org/10.7554/eLife.36440.023
Author response https://doi.org/10.7554/eLife.36440.024

## Additional files
### Supplementary files
• Transparent reporting form
DOI: https://doi.org/10.7554/eLife.36440.021

### Data availability
All data generated or analysed during this study are included in the manuscript and supporting files. Source data files have been provided for Figures 1, 3 and 4 and Figure 2-figure supplement 2.

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
