## [Decision Letter]

Thank you for submitting your article "Ciliary and rhabdomeric photoreceptor-cell circuits form a spectral depth gauge in marine zooplankton" for consideration by *eLife*. Your article has been reviewed by three peer reviewers, and the evaluation has been overseen by a Reviewing Editor and Eve Marder as the Senior Editor. The following individuals involved in review of your submission have agreed to reveal their identity: Claude Desplan (Reviewer #2); Shawn Xu (Reviewer #3).

The reviewers have discussed the reviews with one another and the Reviewing Editor has drafted this decision to help you prepare a revised submission. As you will see from the requested list of revisions below, these are all textual clarifications and no new experiments are requested.

Reviewer #1:

1) In the Introduction, please provide additional details regarding the organization of each eye type by opsin expression as well as the features of each circuit so that a naive reader can interpret how the two circuits are integrated.

2) Please provide additional details regarding how the neural network/connectome is shared between cPRCs and rPRCs. Please explain abbreviations such as RGW, NOS. What is the nature of flask-shaped sensory neurons? Are the six interneurons (IN^preMN^) identically contacted by RGW (or other cells)? What is the nature of MS2 and MS2 and are they identically contacted by RGW? You may wish to include a detailed circuit map as a supplemental figure.

3) Regarding the overall convergence of the two sub-circuits it would be important to know how dominant this interaction through these neurons by RGW is – are these the main partners? What fraction of the input of MS1/2, IN^preMN^ and IN^sn^ are from RGW? Is RGW the only link between the two circuits?

4) The sentence "We did not find any neurons that were postsynaptic to therPRC circuit and presynaptic to the cPRC circuit.", is a strong statement: does this statement include indirect connections via another interneuron(s)?

5) Since MS1/2 are proposed to be mechanosensory neurons it may very well be that the RPC circuit is primarily tuned by mechanosensation and/or that cPRCs modulate mechanosensensory output. Please consider this in the Discussion.

6) A point that needs clarification is the hyperpolarization to UV light (and why the authors do not use a UV laser to stimulate but instead, 405nM) followed by depolarization, and the difference with the 488 stimulation.

7) The responses of RGW and its target appears quite relevant for the logic of what the circuits mediate. It may be worth adding this data (Figure 2—figure supplement 2) into an actual figure. How does hyper- or depolarization translate to synaptic release and activation/inhibition of target cells (e.g. RGW)? How does RGW respond to 488 illumination? Why is 488 excitation only resulting in responses in SN^late^? Please not that these questions do not require additional experiments but you may wish to add in comments in the text.

8) The behavioral analysis is quite intriguing, showing a regulated attraction to avoidance switch in only a few minutes. The antagonistic assay is less clear and would benefit from better description. In this assay several (diffuse lateral and above directional) light sources are combined with different wavelengths. Interestingly in the supplemental video it appears that animals either show positive or negative phototaxis and that the ratio of animals displaying each is altered between time points. Could the authors elaborate on this and how it affects their conclusion?

9) c-opsin1 mutants show an almost complete loss of GCaMP activity in cPRCs. While phototaxis to 480 remains unaffected, the switch to photo-attraction at 395 is lacking in c-opsin1 mutants. However it appears that the switch is not completely absent as animals appear to come to displacement of 0. This would be a distinct from the current notion that the animals do not show UV-avoidance. Please comment on this (also see comment 11 below).

10) Please provide some discussion of the type of neurotransmitters in the photoreceptors and the interneurons that allow the circuit to function properly (since ciliary photoreceptors hyperpolarize to light). The authors mention that neuroendocrine volume transmission is involved in UV avoidance but this needs to be clarified and further discussed.

11) The c-opsin1 mutant data provides strong evidence that UV-evoked downward swimming behavior (UV avoidance) is mediated by cPRC. Since the blue light induced upward swimming behavior is not affected in this mutant, presumably, this mutant should have a defect in vertical swimming direction in the ratio of UV/blue light assay, as shown in Figure 3G for wild-type. However, this experiment is not done. While no new experiments are required, could this issue be addressed in the writing?

---

## [Author Response]

Reviewer #1:1) In the Introduction, please provide additional details regarding the organization of each eye type by opsin expression as well as the features of each circuit so that a naive reader can interpret how the two circuits are integrated.

We have now updated the Introduction and added more detail on photoreceptor organisation, opsin expression and the features of the *Platynereis* eye and cPRC circuits. We also included several references on opsin expression in rPRCs and cPRCs.

2) Please provide additional details regarding how the neural network/connectome is shared between cPRCs and rPRCs. Please explain abbreviations such as RGW, NOS. What is the nature of flask-shaped sensory neurons? Are the six interneurons (IN^preMN^) identically contacted by RGW (or other cells)? What is the nature of MS2 and MS2 and are they identically contacted by RGW? You may wish to include a detailed circuit map as a supplemental figure.

We have provided more information on the connectome. We included a figure supplement (Figure 1—figure supplement 1) with a more detailed circuit map. We also added the complete ungrouped connectivity matrix (Figure 1—source data 2). We explained the abbreviations RGW and NOS. The nature of the flask-shaped sensory cells has been discussed in more detail. These cells were described in (Williams et al. 2017).

We give more detail about the nature of the MS neurons and also cite our recent preprint where more information can be found on these cells.

3) Regarding the overall convergence of the two sub-circuits it would be important to know how dominant this interaction through these neurons by RGW is – are these the main partners? What fraction of the input of MS1/2, IN^preMN^ and IN^sn^ are from RGW? Is RGW the only link between the two circuits?

To address this, we have now added a new figure supplement (Figure 1—figure supplement 2), corresponding to the grouped connectivity matrix in Figure 1, that shows for each connection in the matrix what fraction of inputs that connection provides relative to the total number of inputs (synapses). For MS1/2, IN^preMN^ and IN^sn^ the fraction of inputs from each of the four RGW cells is 25%, 30% and 19%, respectively.

4) The sentence "We did not find any neurons that were postsynaptic to therPRC circuit and presynaptic to the cPRC circuit.", is a strong statement: does this statement include indirect connections via another interneuron(s)?

It also includes indirect connections via maximum one other neuron. We explain now more clearly in the text that we searched the entire head connector graph for paths, using the Catmaid Graph widget (grow paths by 2 hops). This strong statement is based on a comprehensive search for all connections in the complete (but still partly unpublished) head connectome.

5) Since MS1/2 are proposed to be mechanosensory neurons it may very well be that the RPC circuit is primarily tuned by mechanosensation and/or that cPRCs modulate mechanosensensory output. Please consider this in the Discussion.

We now mention this possibility in the Discussion.

6) A point that needs clarification is the hyperpolarization to UV light (and why the authors do not use a UV laser to stimulate but instead, 405nM) followed by depolarization, and the difference with the 488 stimulation.

We used 405 nm light for stimulation during calcium imaging for technical reasons. We only had a 405 nm but not a UV laser line available on the confocal microscope we used for these experiments. Please note that 405 nm is within the absorption range of c-opsin1 (Figure 2E). We now discuss in more detail the hyperpolarisation followed by depolarisation and the difference with the 488 nm stimulation in the Discussion.

7) The responses of RGW and its target appears quite relevant for the logic of what the circuits mediate. It may be worth adding this data (Figure 2—figure supplement 2) into an actual figure. How does hyper- or depolarization translate to synaptic release and activation/inhibition of target cells (e.g. RGW)? How does RGW respond to 488 illumination? Why is 488 excitation only resulting in responses in SN^late^? Please not that these questions do not require additional experiments but you may wish to add in comments in the text.

Examples of RGW-cell responses were already shown in Figure 2J as well as the figure supplement. We have now included more examples and also show that the RGW cells do not respond to 488 nm stimulation. We have also added more examples of SN^early^ and SN^late^ responses. SN^late^ cells do not consistently respond to 488 nm and the observed responses were weaker. We also address in the Discussion how cPRC hyper- and depolarisation could relate to responses in RGW and other cells.

8) The behavioral analysis is quite intriguing, showing a regulated attraction to avoidance switch in only a few minutes. The antagonistic assay is less clear and would benefit from better description. In this assay several (diffuse lateral and above directional) light sources are combined with different wavelengths. Interestingly in the supplemental video it appears that animals either show positive or negative phototaxis and that the ratio of animals displaying each is altered between time points. Could the authors elaborate on this and how it affects their conclusion?

We have now reworded the section describing the antagonistic experiment (Figure 3G). We only used directional light stimulus from above and diffuse infrared light from the side to illuminate the larvae for recording. The supplementary video shows a recording when directional UV stimulation was switched on from above and the larvae displayed positive phototaxis initially, followed by UV avoidance. We have several reasons to believe that this downward swimming is not phototaxis but UV avoidance behavior. We clarified these points in the Discussion.

9) c-opsin1 mutants show an almost complete loss of GCaMP activity in cPRCs. While phototaxis to 480 remains unaffected, the switch to photo-attraction at 395 is lacking in c-opsin1 mutants. However it appears that the switch is not completely absent as animals appear to come to displacement of 0. This would be a distinct from the current notion that the animals do not show UV-avoidance. Please comment on this (also see comment 11 below).

We have changed the text to reflect this. We still use the wording “c-opsin1^Δ8/Δ8^ larvae have a defective UV-avoidance response” since the response is drastically diminished and we do not observe downward swimming in the mutants. We now mention that there may be an alternative UV-avoidance mechanism, responsible for the diminishing phototactic upward swimming in c-opsin1 mutants.

10) Please provide some discussion of the type of neurotransmitters in the photoreceptors and the interneurons that allow the circuit to function properly (since ciliary photoreceptors hyperpolarize to light). The authors mention that neuroendocrine volume transmission is involved in UV avoidance but this needs to be clarified and further discussed.

We have now mentioned the type of neurotransmitters in the different neurons of the circuit, where we know these. The cPRCs, RGW interneurons are peptidergic/cholinergic, the MS cells are cholinergic, the rPRCs are glutamatergic (adult eyes) or cholinergic (eyespot rPRC). We have unpublished genetic evidence that a diffusible neuroendocrine signal is critical for UV-avoidance behavior. We mention this in the Discussion. These data form the basis of an independent project with a manuscript currently in preparation, therefore we prefer not to discuss them in detail here.

11) The c-opsin1 mutant data provides strong evidence that UV-evoked downward swimming behavior (UV avoidance) is mediated by cPRC. Since the blue light induced upward swimming behavior is not affected in this mutant, presumably, this mutant should have a defect in vertical swimming direction in the ratio of UV/blue light assay, as shown in Figure 3G for wild-type. However, this experiment is not done. While no new experiments are required, could this issue be addressed in the writing?

We have addressed this in the Results section. We now write that “The loss of c-opsin1 is expected to disrupt the depth gauge since in c-opsin1 mutants both cyan and UV light induce positive phototactic behavior.” We tested the c-opsin1 mutant under the two extremes of UV/blue ratio, namely 100% UV or 100% blue light. Since both wavelengths induce upward swimming, we did not think it was necessary to try intermediate ratios.